# Co-Precipitated Ni-Mg-Al Hydrotalcite-Derived Catalyst Promoted with Vanadium for CO_2_ Methanation

**DOI:** 10.3390/molecules26216506

**Published:** 2021-10-28

**Authors:** Paulina Summa, Katarzyna Świrk, Dominik Wierzbicki, Monika Motak, Ivo Alxneit, Magnus Rønning, Patrick Da Costa

**Affiliations:** 1Institut Jean Le Rond d’Alembert, Sorbonne Université, CNRS UMR 7190, 78210 Saint-Cyr-L’Ecole, France; 2Faculty of Energy and Fuels, AGH University of Science and Technology, 30-059 Kraków, Poland; dominik.wierzbicki@psi.ch (D.W.); motakm@agh.edu.pl (M.M.); 3Department of Chemical Engineering, Norwegian University of Science and Technology (NTNU), 7491 Trondheim, Norway; katarzyna.swirk@ntnu.no (K.Ś.); magnus.ronning@ntnu.no (M.R.); 4Paul Scherrer Institut (PSI), 5232 Villigen, Switzerland; ivo.alxneit@psi.ch

**Keywords:** methanation, hydrotalcite, vanadium, nickel catalyst, mixed oxides, CO_2_ hydrogenation

## Abstract

Co-precipitated Ni-Mg-Al hydrotalcite-derived catalyst promoted with vanadium were synthesized with different V loadings (0–4 wt%) and studied in CO_2_ methanation. The promotion with V significantly changes textural properties (specific surface area and mesoporosity) and improves the dispersion of nickel. Moreover, the vanadium promotion strongly influences the surface basicity by increasing the total number of basic sites. An optimal loading of 2 wt% leads to the highest activity in CO_2_ methanation, which is directly correlated with specific surface area, as well as the basic properties of the studied catalysts.

## 1. Introduction

The emission of greenhouse gases, especially CO_2_, is an important problem that society is currently facing [1]. Considering the increasing threat related to global warming, such as the intensification of floods/drought periods, heat waves, and temperature rise [2,3], more restrictive politics related to CO_2_ emissions are being introduced. In December 2020, during the EU summit, it was decided to cut 55% of CO_2_ emissions before year 2030 [4]. Therefore, the fully decarbonized global industry is crucial to limit the climate change and to limit the global warming up to 2 °C by 2050–2070 [5]. This will involve an increase in energy efficiency for carbon capture, electrification, and sustainable production of hydrogen [5]. In this regard, carbon dioxide can be utilized in several chemical processes. Technologies such as the production of cyclic carbonates, dimethyl carbonate, and hydrogenation of CO_2_ to methanol have already been developed on an industrial scale [6]. In addition, the CO_2_ hydrogenation process to methane is an emerging technology and has been industrialized by companies such as Audi, DVGW, etc. [7].

The CO_2_ methanation or Sabatier’s reaction is based on the exothermic reaction (1) between carbon dioxide and hydrogen.
CO_2_ + 4H_2_ = CH_4_ + 2H_2_O, ΔH_298K_ = −165 kJ/mol(1)

The products of the reaction are methane and water vapor, making it a solution for both the utilization of CO_2_ and the storage of chemical energy in the form of synthetic gas. 

CO_2_ methanation is a kinetically limited catalytic process, requiring the design of a stable and selective catalyst. A commercially available catalyst used for this process is alumina-supported nickel (Ni/Al_2_O_3_). Still, this material can be improved to ensure prolonged lifetime [8,9]. Ni/Al_2_O_3_ catalysts are prone to sintering of nickel particles on the surface at high temperatures and the deactivation of active centers below 250 °C [10]. Additionally, the activity of the monometallic nickel catalyst is assigned to the concentration of nickel: the higher the content of nickel, the better the activity [11]. These problems can be overcome in several ways, for example, by substituting nickel for another type of active material such as rhodium or ruthenium. The latter metals are more stable and their low loadings is sufficient to provide satisfactory activity. On the other hand, the use of noble metals is expensive and requires special utilization, making the production of such catalysts uneconomical. The activity of the nickel-based catalyst can be improved by changing the synthesis method to one that provides better dispersion of the active phase and improved metal-support interaction, which may limit sintering to a certain extent [12].

An effective synthesis of multielemental catalysts, with dedicated metal-support interaction could be the thermal decomposition of hydrotalcites. Hydrotalcites are layered double hydroxides, with water and ions present in the interlayer region, with the chemical formula presented in Equation (2) [13]. Brucite-like layers in natural hydrotalcite are formed with Mg and Al ions, although during the synthesis of synthetic minerals it is possible to substitute them with other elements of similar atomic radius, which allows the preparation of complex, uniform materials [14]. Above 400 °C, hydrotalcite undergoes thermal decomposition to mixed oxides with a periclase-like crystalline structure. Reduction of this type of oxides leads to the formation of nickel nano-crystallites resistant towards sintering due to the strong metal-support interaction and high dispersion [11].
(2)$$\begin{array}{c} {\left[M_{\left(1-x\right)}^{2+}M_{x}^{3+}{\left(\mathrm{OH}\right)}_{2}\right]}^{x+}(A^{n-}{}_{\frac{x}{n}})\cdot {\mathrm{mH}}_{2}O\\ M^{2+}—\mathrm{bivalent}\;\mathrm{metal}\;\mathrm{ion},\\ M^{3+}—\mathrm{trivalent}\;\mathrm{metal}\;\mathrm{ion},\\ x\;\mathrm{from}\;0.2\;\mathrm{to}\;0.33,\\ A—\mathrm{anion} \end{array}$$

Additionally, choosing a more complex support material may improve its interaction with the active phase and provide better acidic-basic properties [15]. γ-Alumina is inexpensive, provides satisfactory surface properties, and inhibits nickel sintering [16]. In addition, its basic properties can be improved via the addition of stronger centers, such as magnesium or cerium to increase the CO_2_ adsorption capacity [17].

Another promising solution is the promotion of supported nickel catalysts with other types of transition metals. This promotion can prevent the sintering of nickel, improve the dispersion, and can modify the surface properties leading to higher activity in CO_2_ methanation. The most commonly used promotors are copper [18], yttrium [19], or manganese [20]. Additionally, lathanum was found to work as a structural promoter, by limiting the number of side reactions [21,22]. Electronic promoters such as molybdenum [23], iron [20], or cobalt [24] are responsible for increasing the electron density of the catalytic sites, consequently increasing the catalytic activity. Vanadium was found to be a promising catalytic solution for catalytic CO_2_ utilizations [25,26,27,28,29]. For double-layered materials, vanadium can be introduced into the hydrotalcite structure both by interlayer doping or by layer doping. Interlayer doping occurs via incorporation of the vanadate complex to the the brucite-like layers, whereas layer doping is based on the introduction of V^3+^ ions to the brucite layers, partially replacing Al^3+^ [30,31].

When used as promotor, vanadium is recognized as a potential CO_2_ methanation catalyst. Thus vanadium was found to increase Ni dispersion, enhance H_2_ uptake, and additionally, VOx provides an electronic effect that can promote the dissociation of CO in the methanation reaction [26,27,28,29]. In our previous study, Mg-Al-V hydrotalcites with a vanadium loading from 0 to 4 wt% were impregnated with nickel and subsequently used as catalysts for CO_2_ methanation. The obtained results confirmed the promotion effect of vanadium throughout the increase of the Ni dispersion, the increase of specific surface area, and an optimal content of weak and medium strength basic sites [32].

In the present study, the preparation method was modified to a one-step synthesis. The active phase was synthesized simultaneously with the support during the co-precipitation at constant pH. This synthesis approach may allow better availability of vanadium on the catalyst surface and thus lead to high catalytic activity.

## 2. Results

### 2.1. Catalytic Tests

#### 2.1.1. Temperature Programmed Surface Reaction

Samples were tested in the CO_2_ methanation reaction as a function of temperature. Results are presented in Figure 1a. Among the tested materials, the highest catalytic activity was demonstrated by a sample containing 2 wt% vanadium. The Ni-CP-V2.0 catalyst not only reached satisfactory activity at temperatures higher than 350 °C, but was also significantly active at 250 °C compared to other materials. The samples containing 0, 1, and 4 wt% of vanadium showed no significant activity below 350 °C. Among them, the most active was the unpromoted catalyst (Ni-CP-V0.0). Ni-CP-V1.0 resulted in slightly better catalytic performance than Ni-CP-V4.0, which may suggest that the increase of vanadium loading is not favorable. This statement can be confirmed by the selectivity of the tested catalysts at the temperature of 300 °C (Figure 1b). The only sample that resulted in the near-equilibrium (99–98%) selectivity to methane was the 2 wt% V promoted sample. The unpromoted catalyst operated with a high selectivity of ca. 95% during the entire test.

For the catalysts containing 1 and 4 wt% vanadium, the selectivity at low temperatures was rather related to the formation of carbon monoxide. Ni-CP-V4.0 was more selective towards CO than Ni-CP-V1.0, although both of them mainly promoted the reverse water-gas shift reaction (RWGS). This is in good agreement with the study of Le et al., which confirmed that the main product of the reaction is carbon monoxide, formed mainly in the RWGS reaction [33]. A similar effect was observed for other types of promoters such as copper, especially promoted on hydrotalcite-derived catalysts [18].

#### 2.1.2. Stability Tests at 300 °C

Steady-state tests were carried out at 300 °C for 5 h for all studied catalyst at the same operating conditions and presented in Figure 2. A similar trend in activity at 300 °C was observed when compared to that recorded during the TPSR test. The conversions for Ni-CP-V0.0, Ni-CP-V1.0, Ni-CP-V2.0, and Ni-CP-V4.0 were slightly higher (ca. 10%) and stable during the entire 5 h test. The CH_4_ selectivity was found to be stable and higher than 91% for both Ni-CP-V0.0 and Ni-CP-V1.0. However, the methane selectivity for Ni-CP-V4.0 decreased from 84 to 79%, showing a possible negative impact of high V loading, which is favorable for CO production [33]. Finally, Ni-CP-V2.0 showed high conversion and high methane selectivity during time on stream.

Additionally, the dispersion (D) and the turnover frequency (TOF) were calculated to measure the efficiency of studied catalysts (the procedure presented in SI). Calculations are based on the performance registered during the stability test at 300 °C. Among the studied catalysts, Ni-CP-V2.0 turned out to be the most efficient material with the highest value of TOF (7.9×10^−2^ s^−1^) (Table 1). The catalysts containing 1 and 4 wt% vanadium resulted in the same dispersion value, and TOF of 3.5 and 3.1×10^−2^ s^−1^, respectively. Ni-CP-V0.0 presented the lowest turnover frequency, but higher dispersion. On the other hand, Ni-CP-V2.0 showed considerably higher TOF values compared to other promoted hydrotalcite-derived catalysts reported in the literature. Summa et al. studied promotion of Cu under the same conditions of CO_2_ methanation (300 °C and GHSV of 12,000 h^−1^). In this study the best performing Cu-promoted catalyst (20 wt% Ni and 1 wt% Cu) revealed TOF of 5.91×10^−2^ s^−1^ [18]. Another example can be La-promoted hydrotalcite-based catalyst containing 40 wt% of nickel and 2 wt% of lanthanum. In the study of Wierzbicki et al., TOF was found to be equal 3.4×10^−2^ s^−1^ (250 °C and GHSV of 12,000 h^−1^) [22].

In order to understand these trends, the following sections present the physicochemical characterizations of the tested catalysts.

### 2.2. Physico-Chemical Properties of V-Promoted Catalysts

#### 2.2.1. Catalyst Reducibility

H_2_-TPR measurements were performed in order to investigate the reducibility of the studied hydrotalcite-derived samples. For all tested samples, a broad peak was observed at the high temperature region (740–850 °C) (Figure 3).

This peak can be assigned to the reduction of the nickel present in the periclase-like mixed-oxide structure. This typical reaction temperature peak at ca. 850 °C corresponds to the reduction of ca. 20 wt% of nickel in the Ni/Mg/Al matrix [11,34]. This later peak is shifted to lower temperatures, with the increase of vanadium content, with a maximum at 740 °C for Ni-CP-V4.0. This shift can be explained by the weakening of the interaction between nickel and the oxide matrix, under the influence of vanadium. In our previous study, the reduction of impregnated V from Mg-Al hydrotalcite-derived oxide matrix was investigated (Appendix A). The collected profiles showed a narrow peak at 590–610 °C, corresponding to the reduction of vanadium from V^5+^ to V^4+^ [35]. Since V^5+^ is reduced in the temperature range of the measurement, it is possible to assume that it ungergoes simulateous reduction with nickel [36].

For V-promoted catalysts, additional low-temperature peaks with two maximums are detected. A broad and intense peak was observed for the Ni-CP-V1.0 sample with a shoulder at 400 °C, and a sharp peak with maximum at 450 °C. According to the literature, these peaks correspond to the reduction of weakly bonded nickel oxides. The intensity of these peaks obtained at low-temperature region decreases with the loading of vanadium. Additionally, the reduction temperature of these peaks increased slightly, ca. 10–20 °C, suggesting stronger interaction of surface NiO with the support.

#### 2.2.2. Low-Temperature N_2_ Sorption on Studied Catalysts

Low-temperature N_2_ sorption measurements were performed to investigate the structure properties of the examined samples. All materials resulted in an isotherm with a well-defined hysteresis loop typical for the type IV of isotherm, classified additionally as a subtype H1, typical for mesoporous materials (Appendix A). H1 subtype suggests the presence of cylindrical-like pore channels or agglomerates of approximately uniform spheres [37]. Hydrotalcite-derived mixed oxides usually present such a type of surface structure, which is compared in Table 2 [18]. The specific surface area is 179 and 207 m^2^/g for the samples with 1 and 4 wt% of vanadium, respectively. Additionally, the mesopore volume of those materials was ca. 20 cm^3^/g. The highest S_BET_ of 306 m^2^/g was obtained with the Ni-CP-V2.0 catalyst. This catalyst additionally resulted in the highest volume of mesopores (0.70 cm^3^/g). The difference in the surface properties is significant and can be assigned as one of the most important factors related to the catalytic activity, since a well-developed surface allows for a better dispersion of active sites.

The unpromoted sample presented a specific surface of 241 m^2^/g, with a mesopore volume of 0.42 cm^3^/g. Mean pore diameter for all hydrotalcite-derived oxides was similar with ca. 10 nm. It is generally assumed that an increase of the promoter loading in hydrotalcite-based catalyst can lead to a decrease of the specific surface area due to the blockage of mesopores [38,39].

#### 2.2.3. Evolution the Crystalline Structure of the Catalysts

To investigate the evolution of the crystalline composition of V-promoted hydrotalcites, XRD was performed on both calcined and reduced samples (Figure 4a,b). The thermal decomposition of hydrotalcite is temperature sensitive. Above 300 °C, the structure of Mg-Al hydrotalciteis is replaced with an oxide similar to periclase [40]. The crystallinity of the resulting phase increases with increasing temperature of calcination. It has been reported that additionally above 600–700 °C, spinel-type oxide may form in thermally decomposed hydrotalcites [40,41]. During calcination, the structure of hydrotalcite was broken down into mixed oxides similar to periclase. All tested samples showed five intense reflexions at 2θ of 37.32 (111), 43.36 (020), 62.99 (022), 75.56 (131), and 79.57 (222), confirming the expected periclase (ICDD 01-087-0653). The scattering characteristic of the amorphous phase was not detected, the peaks were uniform and broad, suggesting a small crystallite size and a decrease in crystallinity. Among the V-promoted samples, a residual phase of hydrotalcite was detected in all materials, suggesting no complete thermal decomposition of the parent structure. It may suggest that the presence of vanadium leads to an increase in the temperature of thermal decomposition of hydrotalcite, most likely throughout increasing the strength of coordination binding in brucite-like layers, requiring more energy to remove -OH groups.

On reduced catalysts, three phases were distinguished from X-ray diffractograms, such as periclase (ICDD 00-045-0946) already discussed in Figure 4a, spinel (ICDD 01-070-5187), and metallic nickel (ICDD 03-065-0380). The source of the emerging nickel phase was the reduction of nickel species present in the periclase-like structure and a such crystalline phase was confirmed with three sharp and narrow reflections at 2θ of 44.3 (111), 51.7 (200), 76.1 (220), typical for metallic nickel. Spinel-type oxide is usually formed at high-temperatures, below 900 °C, at which the reduction was carried out, the temperature was sufficiently high to form such structure. This phase was most likely formed from periclase-like oxide and the obtained spinel oxide was MgAl_2_O_4_, as confirmed by the reflections at 2θ of 19.0 deg (111), 31.3 deg (220), 36.8 deg (311), 44.8 deg (400), 59.3 deg (511), and 65.2 deg (440). Additionally, a certain difference in the intensity in the diffractograms of the samples was visible: Ni-CP-V0.0 and Ni-CP-V2.0 resulted in much lower intensity of the nickel-related reflections, suggesting a small nize of the nickel crystallites in the sample. For Ni-CP-V1.0 and Ni-CP-V4.0, the reflections assigned to nickel are very intense, narrow, and sharp, indicating a relatively large size of metal crystallites. After the reduction step, no additional peaks related to vanadium oxide or other vanadium species were detected. Additionally, there is no shift in the position of the reflexions characteristic to nickel, suggesting that the Ni-V solid solution is not formed.

Based on Rietveld refinements, the fraction of each of the detected crystalline phases was calculated, based on the height, width, and position of the reflexions. Results are compared in Table 3. Several trends were possible to distinguish. The content of spinel was increased with the amount of vanadium promotion, from 28.6 for Ni-CP-V0.0 up to 50.1% for Ni-CP-V4.0. On the contrary, with the increase of vanadium content, the share of periclase-like oxide structure was decreased from 63.1% for the unpromoted catalyst to 23.5% for the sample with the highest loading of vanadium. The content of metallic nickel phase was of 18–19% for Ni-CP-V0.0 and V2.0, while V1.0 and V4.0 catalysts resulted in ca. 26%. Additionally, crystallite size based on the Scherrer’s equation was calculated for nickel. The smallest crystallites were found in the Ni-CP-V2.0 catalyst—with a size of ca. 6 nm. Larger crystallites were found for the unpromoted sample, of ca. 15 nm. Nickel particles of 31 and 33 nm were found for Ni-CP-V4.0 and Ni-CP-V1.0 samples. The samples with the larger nickel particle size were those with a higher content of nickel detected on the surface. It may suggest that during the reduction step formation of bigger crystallites was favoured due to the higher amount of nickel available on the surface.

#### 2.2.4. Surface Basicity of the Reduced Catalysts

CO_2_-TPD was performed to evaluate the number of basic sites available on the surface of the reduced hydrotalcite-derived catalysts. CO_2_-TPD profiles (Figure 5) were deconvoluted into three gaussian curves with maxima at the ranges 125–150 °C, 185–220 °C, and 270–340 °C, corresponding to weak, medium-strenght, and strong basic sites, subsequently. The values corresponding to distribution of each type of sites are compared in Table 2. The lower number of total basic sites was found for the non-promoted catalyst Ni-CP-V0.0. With the increase of vanadium content, the amount of basic sites significantly increases. Promotion with 1 wt% of V resulted in doubling the number of total basic sites, confirming the strong basic properties of vanadium.

In the studied V-promoted catalysts, medium strength basic sites, which have been reported to play an important role in the CO_2_ methanation reaction [22], were the most abundant. For the unpromoted hydrotalcite-derived catalyst, the distribution of the types of basic sites is uniform, and no type of sites were found to be dominating. Additionally, with the increase of vanadium promotion, the number of strong basic sites assigned to Lewis basic sites associated with oxygen anions increased [42]. The main role assigned to the basic sites is improvement of the chemisorption of CO_2_ on the support and prevention of cokeformation on the catalyst surface [43].

During methanation reaction, CO_2_ forms monodentate carbonates on basic low-coordination oxygen anions that are present at the surface, especially in MgO [18]. The temperature of CO_2_ desorption from the strong sites is decreased for V-containing materials, which suggests a weakened interaction of unidenate ions in comparison to the unpromoted material, or lower availability of MgO on the surface. Ni-CP-V2.0 is the sample with the highest number of weak basic sites assigned to Brønsted basic centers, such as hydroxyl groups where bicarbonate is formed (Table 4). Ni-CP-V4.0 was found to be dominating in the case of medium strength basic sites characterized as Lewis acid base sites, where CO_2_ forms bidentate carbonates on metal-oxygen pairs. For Ni-CP-V0.0 and Ni-CP-V2.0 low and moderate temperature peaks were shifted ca. 20 °C to higher temperatures, suggesting stronger interactions between CO_2_ and sites in those two samples.

#### 2.2.5. Morphology and Structure of Catalysts by Transmission Electron Microscopy

Figure 6 shows HRTEM images for Ni-CP-V0.0, (Figure 6(1A,1B)), Ni-CP-V1.0 (Figure 6(2A,2B), Ni-CP-V2.0 (Figure 6(3A,3B), and Ni-CP-V4.0 (Figure 6(4A,4B), with histograms (bin width calculated by the rice-estimator [44]) representing the distribution of nickel particle sizes. From Figure 6, one can conclude that the distribution of the nickel particles is uniform in all the hydrotalcite-derived samples.

The Ni^0^ average particle size were found as follow: 12, 11, 15, and 28 for Ni-CP-V0.0, Ni-CP-V1.0, Ni-CP-V2.0, and Ni-CP-V4.0, respectively (Figure 6(C)). Furthermore, the results obtained by TEM suggest that the size of nickel crystallites increased with the increase of vanadium loading, which is not in line with the XRD results. On the contrary, in the literature, on V-Mg-Al mixed oxides impregnated with nickel, it was found that optimal vanadium loading (2 wt%) led to lower Ni^0^ particle size [32].

Additionally, energy dispersive X-ray (EDX) mapping analysis were performed on V-promoted samples (Ni-CP-V2.0 and Ni-CP-V4.0), to investigate the distribution of Ni and V on the sample (full EDX analyses are presented in Appendix A). Nickel particles appeared to be regularly shaped spherical crystallites. Vanadium was found to be homogenously dispersed in the support and no enrichment was found at the positions of the Ni particles. This seems to exclude the formation of solid V/Ni solutions with a significant vanadium content even when the V loadings increased as reported elsewhere [45]. 

### 2.3. Post-Run Characterizations

XRD was carried out on post-run samples to investigate the possible changes in the crystalline structure of the tested hydrotalcite-derived catalysts and a possible sintering of Ni particles. For all samples, three types of a crystalline phase were detected. Those phases were: periclase (ICDD 01-087-0653), spinel (ICDD 01-070-5187), and metallic nickel (ICDD 03-065-0380), similarly to the reduced materials. Additionally, in the non-promoted catalyst, Ni-CP-V0.0, residual hydrotalcite phase was registered. No scattering characteristic for amorphous phases was detected in the diffractograms, suggesting either lack of amorphous phases or presence in the amounts below the possibility of detection with this method. 

The intensity of reflexions from periclase and spinel-like oxides is lower compared to the intensity of the Ni^0^ phase. This observation has already been described in the literature as the lack of stabilization of NiO particles on the surface. This phenomenon is observed when the nickel particles are not stabilized in the Mg-Al oxide matrix and tend to migrate to the surface to sinter [46].

The nickel crystallites size was calculated based on Scherrer’s equation (Figure 7). It was observed, that the large crystallites observed in reduced samples Ni-CP-V1.0 and Ni-CP-V4.0 before the methanation tests were redispersed to smaller ones, with a decrease in the average diameter of ca. 16 and 15 nm. Moreover, the size of nickel crystallites on Ni-CP-V2.0 decreased, from 6 to 4 nm. For unpromoted catalyst, after the catalytic test, a slight increase of the Ni° particle was observed (from 15 to 17 nm), suggesting a slight sintering. Since a decrease in nickel crystallite size was observed for all the V-promoted catalysts, it is possible to conclude that presence of vanadium modifies the surface properties, which leads to a redispersion of Ni° on the catalyst. Similar behaviour was already observed for Cu-Ni mixed oxides derived from hydrotalcites, in which a Cu-Ni alloy could be formed. Such a redispersion was explained due to the susceptibility of nickel atoms to segregate from the solid solution more easily than copper [18,47]. Thus, It is possible to expect a similar effect either on vanadium promoted catalysts, although, EDX mapping confirmed lack of formation of Ni-V alloy on the surface, so this behaviour could be assigned to the oxygen storage capacity of the V promoted catalyst, which can be intensified in this case by the electronic properties of V [48].

TEM image of spent Ni-CP-V2.0 catalyst (Figure 8) revealed small nickel crystallites, with an average diameter of ca. 10 nm, whereas 15 nm were found on the reduced sample. This decrease confirms the tendency found previously in XRD. Thus, in our study it is possible to exclude the sintering of nickel. These later results confirmed the redistribution of nickel particle after methanation reaction. Moreover, one can note that the distribution of nickel in post-run sample was regular and similar to what was visible on the reduced catalysts before the test. Based on EDX mapping, no agglomeration of vanadium particles was detected—it seems that they remained in the Mg-Al oxide matrix, forming a homogenous solid solution (full EDX analyses are presented in Appendix A).

## 3. Correlation between the Catalytic Activity and the Physico-Chemical Properties of the Ni-CP-VX Catalysts

Based on the results of the CO_2_ methanation tests, and physico-chemical characterisation, it is possible to draw a correlation between the catalytic activity and the properties of the V-promoted nickel catalyst (Figure 9). The activity of CO_2_ conversion at 350 °C was correlated with Ni^0^ crystallite size, total number of basic sites, and specific surface area determined on the samples. A direct correlation was found between the CO_2_ conversion, specific surface area and Ni^0^ crystallite size. Among the studied materials, Ni-CP-V2.0 (72% at 350 °C) was found to be the best catalyst. This one possesses the higher specific surface area of 306 m^2^/g and the smaller nickel particles (6 nm). The ranking in terms of catalytic versus SSA and Ni° particle size is then as follows: Ni-CP-V0.0. with a CO_2_ conversion of 40% at 350 °C, a SSA of 241 m^2^/g, and Ni^0^ particle size of ca. 15 nm > Ni-CP-V1.0 (CO_2_ conversion = 25%, S_BET_ = 179 m^2^/g, d_Ni_° = 33 nm) > Ni-CP-V4.0 (CO_2_ conversion = 15%, S_BET_ = 207 m^2^/g, d_Ni_° = 31 nm). Thus as expected, parameters such as specific surface area and nickel particle size are correlated because an increase of the surface allows for a better dispersion of Ni^0^. As reported in the CO_2_ TPD results, the promotion with vanadium led to one important increase of the surface basicity. Thus, the V-promoted materials resulted in a higher number of medium-strength and strong basic sites than on the unpromoted catalyst. Additionally, with the increase of vanadium content, the amount of total basic sites was increased as well. It was already reported in the literature that the increase of medium-strength basicity is related to an improvement of the catalytic activity [21]. It is worth to note that for the co-precipitated, V-promoted Ni-Mg-Al hydrotalcite-derived catalysts, despite the significant improvement of the basic properties, especially in terms of medium-strength and strong basicity, there is no correlation between this property and CO_2_ conversion. Most likely, basicity was not the predominant factor here, in comparison to strongly varying specific surface area and dispersion.

## 4. Materials and Methods

Samples were prepared via co-precipitation at constant pH, as proposed by Cavani et al. [13]. Mg(NO_3_)_2_·6H_2_O, Al(NO_3_)_3_·9H_2_O, Ni(NO_3_)_2_·6H_2_O and VCl_3_ were utilized for the synthesis. Samples were co-precipitated at a temperature of 65 °C, stabilizing the pH at the range 9.5–10 by adding 1M NaOH dropwise. The precipitate was aged in mother solution for 24 h at 65 °C, later washed with distilled water and dried overnight in static air at 80 °C. Dry hydrotalcites were calcined for 5 h at 500 °C and reduced prior to the menthanation tests for 1 h at 900 °C in the 5%H_2_/Ar.

Catalysts were characterized by XRD, H_2_-TPR, CO_2_-TPD, low-temperature N_2_ sorption, and TEM. X-ray diffraction (XRD) were performed after calcination, with reduction at the post-reaction stage.

XRD patterns were obtained using a Panalytical Empyrean ((Malvern Panalytical B.V., Almelo, The Netherlands), diffractometer equipped with CuKα (λ = 1.5406 Å) radiation, working in Bragg-Brentano θ-θ geometry. Data was collected within a 2θ range of 3–90 deg. Ni crystallite size was calculated with Scherrer’s equation, with a correction related to the instrument broadening included. Textural properties of the catalysts, specific surface area, and total pore volume were obtained via low-temperature nitrogen sorption. The measurements were carried out on Belsorp Mini II apparatus (BEL Japan, Inc., Osaka, Japan). The samples were outgassed for 2 h at 350 °C beforehand. Both temperature-programmed reduction (H_2_-TPR) and temperature-programmed desorption (CO_2_-TPD) profiles were obtained using a BELCAT-M instrument equipped with a TCD detector (BEL Japan, Inc., Osaka, Japan). Prior to TPR, the calcined sample was outgassed for 2 h at 100 °C and then reduced at the heating ramp of 10 °C/min, from 100 to 900 °C with a gas mixture containing 5% H_2_ in Ar. TPD was carried on the reduced sample, which was first degassed for 2 h at 500 °C and then cooled to 80 °C. After the pretreatment, a mixture of 10% CO_2_/He was fed for 1 h in order to adsorb CO_2_ on the sample, and then pure He was flowed over the sample for 15 min to remove weakly adsorbed carbon dioxide. The temperature range of TPD measurements was from 100 to 800 °C with a heating rate of 10 °C/min. Thermogravimetric analysis (TGA) was performed on TA Q5000 IR thermobalance (TA Instruments, Warsaw, Poland), in the temperature range from 30 to 800 °C. In a typical experiment, ca. 30–45 mg of the material was placed on the Pt holder and equilibrated at 30 °C for several minutes. A heating rate of 10 °C min^−1^ was used, with 100 ml·min^−1^ synthetic air flow provided in the direct vicinity of the sample. A buoyancy effect was found to be negligible.

Transition Electron Microscopy (TEM) was performed on a probe corrected JEOL JEM ARM-200F (NeoARM) (JEOL, Gmbh, Freising, Germany) equipped with a cold FEG gun operated at 200 keV. EDS spectra were collected by two SDD detectors covering 1.7 sr.

A catalytic test was performed inside a tubular fixed-bed quartz U-type reactor heated by the vertical electric furnace. A K-type thermocouple was placed outside the catalytic bed to control the temperature. The inlet gas composition was CO_2_/H_2_/Ar = 1.5/6/2.5 with a flow of 100 ml/min (GHSV = 12,000 h^−1^). The products of the reaction such as CO_2_, CO, CH_4_, and H_2_, were analyzed with an online micro-chromatograph (Varian GC4900) (Agilent Technologies France, Les Ulis, France) equipped with a thermal conductivity detector (TCD). The tests were carried out in the temperature range from 250 °C to 450 °C, with the sample kept for 30 min at each temperature to obtain the steady state. The heating rate between steps was 10 °C/min. Materials were reduced for 1 h at 900 °C in 5%H_2_/Ar, before the reaction. The stability test was performed at 300 °C for 5 h, under similar experimental conditions.

CO_2_ conversion and CH_4_ selectivity were calculated using the Equations (3) and (4):(3)$${\mathrm{CO}}_{2}\;\mathrm{conversion}\;\mathrm{as}\;\chi_{{\mathrm{CO}}_{2}}\left(\%\right)=\frac{F_{{\mathrm{CO}}_{2}\mathrm{inlet}}-F_{{\mathrm{CO}}_{2}\mathrm{outlet}}}{F_{{\mathrm{CO}}_{2}\mathrm{inlet}}}\;\cdot 100,$$
(4)$${\mathrm{CH}}_{4}{\;\mathrm{selectivity}\;\mathrm{as}\;S}_{{\mathrm{CH}}_{4}}\left(\%\right)=\frac{F_{{\mathrm{CH}}_{4}\mathrm{outlet}}}{F_{{\mathrm{CH}}_{4}\mathrm{outlet}}+{\;F}_{\mathrm{COoutlet}}}\cdot 100,$$
where F is the flow calculated from the concentration of the gases.

## 5. Conclusions

The study on vanadium promotion on hydrotalcite-derived Ni-Mg-Al catalyst for CO_2_ methanation confirmes that vanadium plays a role both as a textural and electronic promoter. The addition of V significantly changes the surface properties, such as specific surface area, mesoporosity, and also improves nickel dispersion. On the other hand, the addition of increased amounts of vanadium decreases catalytic activity, lowers selectivity to methane, and leads to the promotion of the reverse water-gas shift reaction. Vanadium additionally strongly influences the surface basicity, increasing the number of basic sites detected on the surface, with increasing V content in the samples. Finally, the V-promoted catalysts prepared by co-precipitation exhibited higher basicity and specific surface area, depending on the V-loading. The co-precipitation method resulted in the formation of an uniform hydrotalcite structure, which was later decomposed into mixed oxides during calcination. This homogeneous matrix of mixed oxides appared to lead to a better distribution of nickel on the surface, an enhanced basicity, and a larger specific surface area during the reduction and subsequent reaction. For comparison, nickel oxide was deposited as a separate phase on Mg-Al-V hydrotalcites in the impregnated materials. This resulted in a lower specific surface area, lower basicity (most likely due to the blockage of Mg-O sites) and ultimately comparable Ni° crystallite size on the surface.

## Figures and Tables

**Figure 1 molecules-26-06506-f001:**
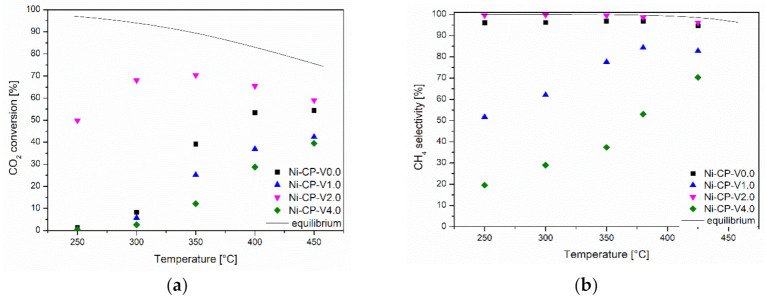
(**a**) CO_2_ conversion as a function of temperature; (**b**) CH_4_ selectivity as a function of temperature for hydrotalcite-derived catalysts.

**Figure 2 molecules-26-06506-f002:**
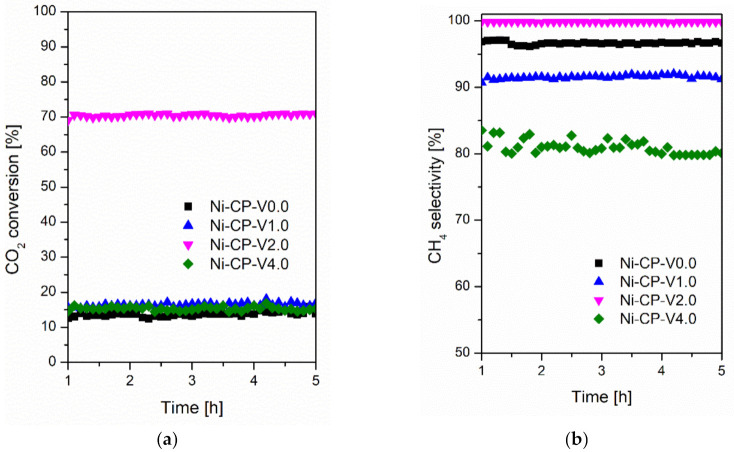
(**a**) CO_2_ conversion as a function of time at 300 °C; (**b**) CH_4_ selectivity as a function of time at 300 °C for hydrotalcite-based catalysts.

**Figure 3 molecules-26-06506-f003:**
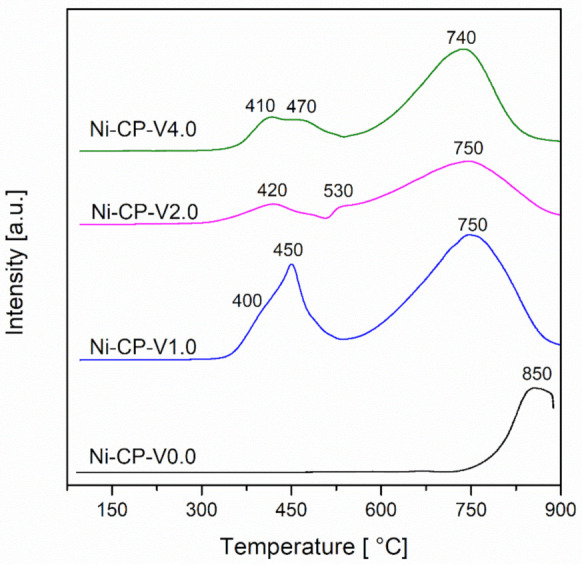
H_2_-TPR profiles for calcined hydrotalcites.

**Figure 4 molecules-26-06506-f004:**
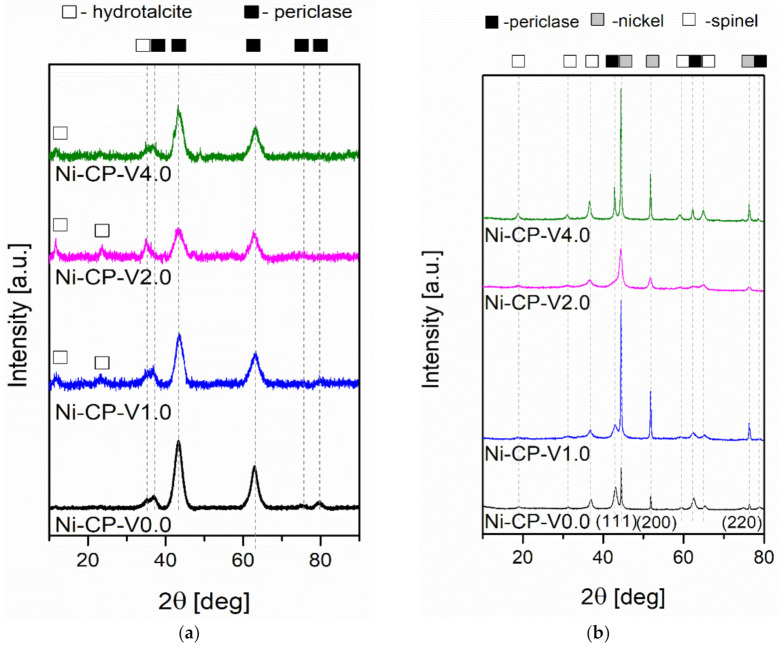
XRD diffractograms for (**a**) catalysts after calcination; (**b**) catalysts after reduction.

**Figure 5 molecules-26-06506-f005:**
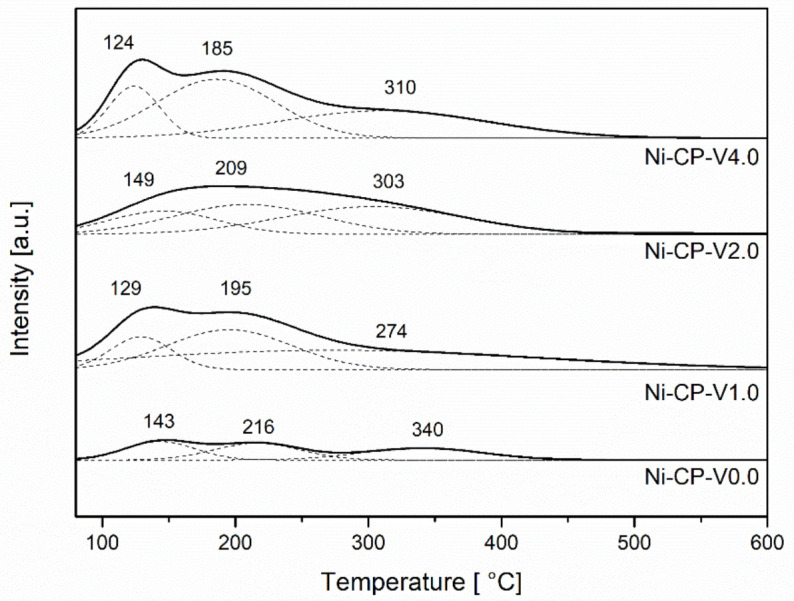
CO_2_-TPD profiles for reduced hydrotalcites.

**Figure 6 molecules-26-06506-f006:**
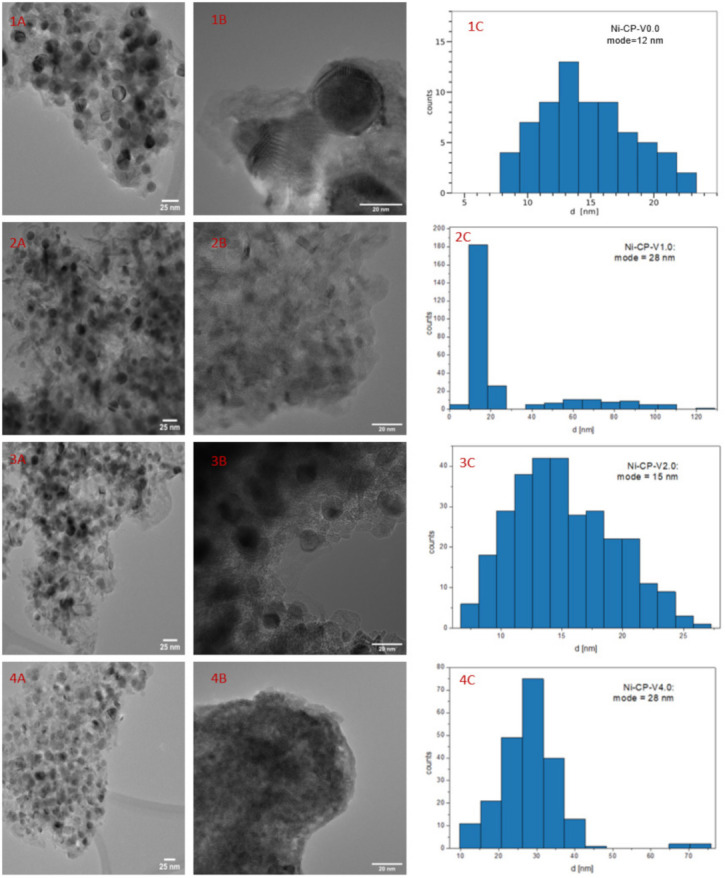
1–4 (**A**,**B**) TEM images for the reduced hydrotalcite-derived samples; 1–4 (**C**) histograms for Ni^0^ particle diameter.

**Figure 7 molecules-26-06506-f007:**
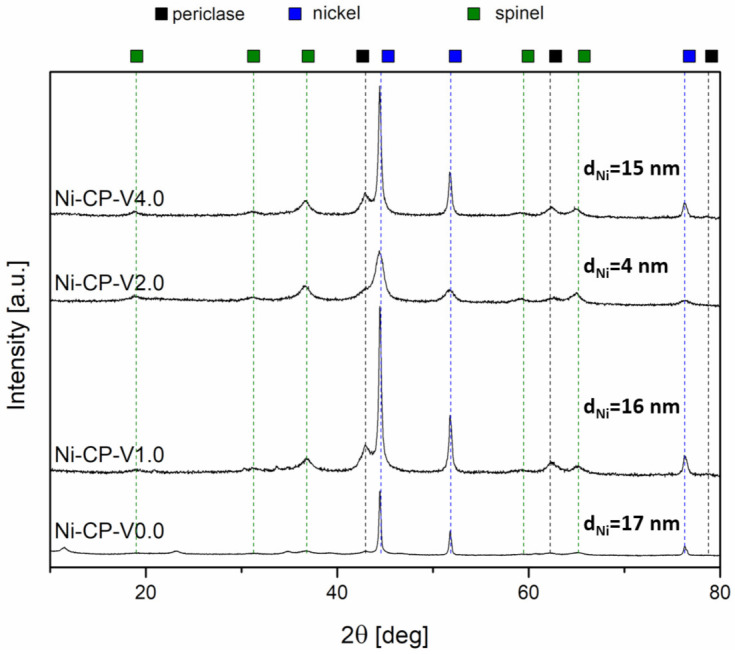
XRD diffractograms for spent hydrotalcite-derived catalyst.

**Figure 8 molecules-26-06506-f008:**
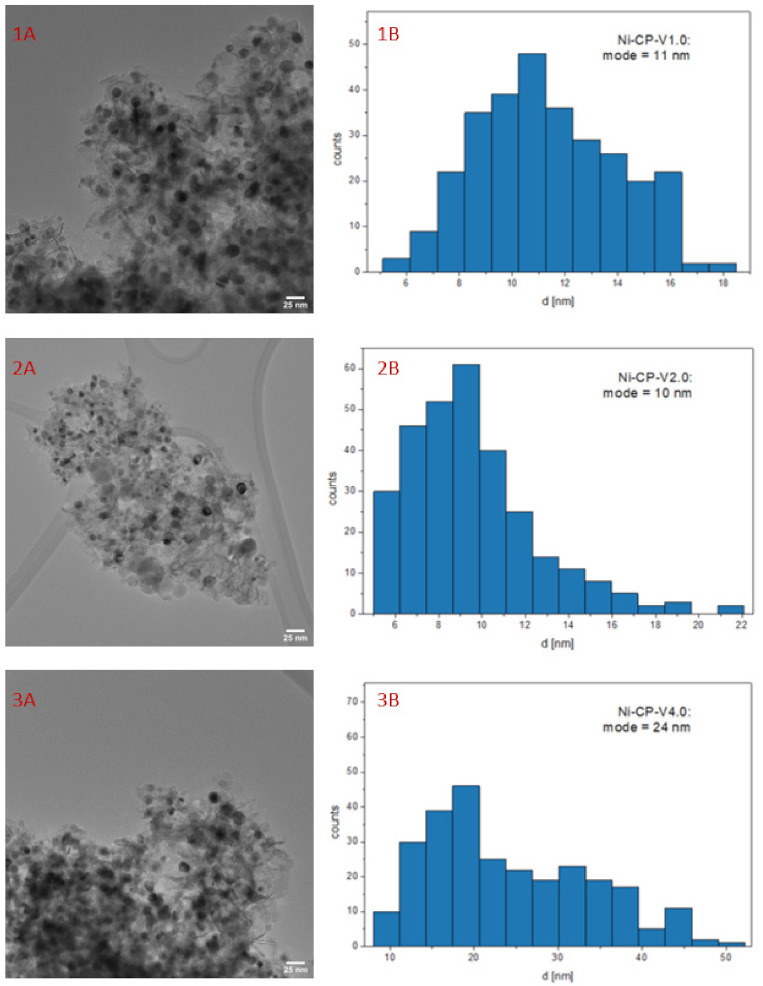
1–3 (**A**) TEM image of spent V-promoted catalysts; 1–3 (**B**) histogram for spent Ni-CP-V2.0.

**Figure 9 molecules-26-06506-f009:**
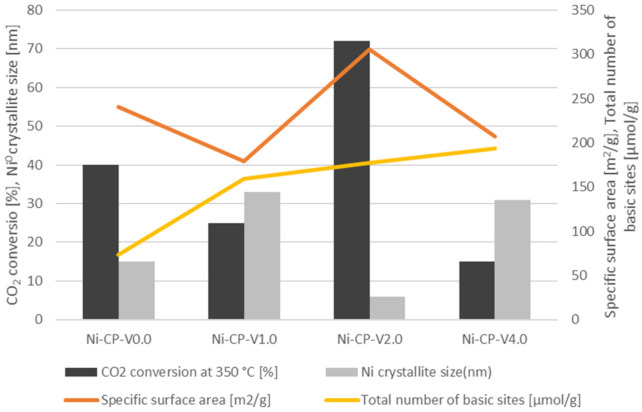
Corellation between vanadium content, CO_2_ conversion, Ni^0^ crystallite size for reduced samples (XRD), specific surface area [m^2^/g], and total number of basic sites [μmol/g].

**Table 1 molecules-26-06506-t001:** Turnover frequency and dispersion for hydrotalcite-derived catalysts.

Sample	Ni-CP-V0.0	Ni-CP-V1.0	Ni-CP-V2.0	Ni-CP-V4.0
TOF [10^−2^ s^−1^]	1.2	3.5	7.9	3.1
D	0.08	0.04	0.07	0.04

**Table 2 molecules-26-06506-t002:** Specific surface area, mean pore diameter, and volume of mesopores of the calcined catalysts.

Sample	S_BET_ [m^2^/g]	Mean Pore Diameter [nm]	Mesopore Volume [cm^3^/g]	Isotherm Type
Ni-CP-V0.0	241	10	0.42	IV—H1
Ni-CP-V1.0	179	10	0.18	IV—H1
Ni-CP-V2.0	306	11	0.70	IV—H1
Ni-CP-V4.0	207	10	0.22	IV—H1

**Table 3 molecules-26-06506-t003:** Results of Rieteveld refinements and Ni^0^ crystallite size calculated based on Scherrer’s equation, for reduced hydrotalcite-derived samples.

Sample	Share of Crystalline Phase			Ni^0^ Crystallite Size [nm]	
	Spinel [%]	Periclase [%]	Nickel [%]	XRD	TEM
Ni-CP-V0.0	28.6	52.9	18.5	15	12
Ni-CP-V1.0	36.2	36.8	26.9	33	28
Ni-CP-V2.0	42	38.9	19.1	6	15
Ni-CP-V4.0	50.1	23.5	26.3	31	28

**Table 4 molecules-26-06506-t004:** Basicity (from CO_2_-TPD) of reduced samples.

Sample	Basic Sites [μmol/g]				Basic Sites Distribution [%]		
	Weak	Medium	Strong	Total	Weak	Medium	Strong
Ni-CP-V0.0	23	25	25	73	32	34	34
Ni-CP-V1.0	36	86	37	159	23	54	23
Ni-CP-V2.0	51	71	55	177	29	40	31
Ni-CP-V4.0	37	98	59	194	19	51	30

## Data Availability

Data is contained within the article.

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
