# Peer review of "Co-Precipitated Ni-Mg-Al Hydrotalcite-Derived Catalyst Promoted with Vanadium for CO2 Methanation"

_molecules, 2021, doi:10.3390/molecules26216506_

Round 1
Reviewer 1 Report
In this manuscript, the authors described the effect of V-doping to Ni-Mg-Al hydrotalcite-derived catalyst on CO2 methanation. Various spectroscopic analysis were carried out to explain the promotion effect of vanadium. The reviewer thinks that the manuscript is potentially publishable in Molecules after addressing the following comments.
- It is unclear whether this V-promoted catalyst is superior to the reported catalysts. The catalytic performance, i.e. TOF or TON of the present catalyst should be compared with those of the reported Ni based catalyst.
- The authors described that the promotion effect of V was ascribed to the change of surface properties and the structure of the calcined catalysts. Does this phenomena appear only when addition of Vanadium?
- In Figure 5 and Table 2, what is the origin of basicity for this catalyst? The authors should explain what is the base sites
- The structure of the obtained catalyst was not shown. The reviewer recommend to draw the model of the catalyst structure after calcination.
Author Response
In this manuscript, the authors described the effect of V-doping to Ni-Mg-Al hydrotalcite-derived catalyst on CO2 methanation. Various spectroscopic analysis were carried out to explain the promotion effect of vanadium. The reviewer thinks that the manuscript is potentially publishable in Molecules after addressing the following comments.
- It is unclear whether this V-promoted catalyst is superior to the reported catalysts. The catalytic performance, i.e. TOF or TON of the present catalyst should be compared with those of the reported Ni based catalyst.
Answer: Thank you for this comment. Calculations of TOF were added and compared to the other catalysts reported in the literature. (Line 148 to 160). A table was added also to the manuscript (Table 1)
- The authors described that the promotion effect of V was ascribed to the change of surface properties and the structure of the calcined catalysts. Does this phenomena appear only when addition of Vanadium?
Answer: Each of the promoters has different effect on the structure of hydrotalcite. Vanadium is foremost textural and electronic promoter, but other elements may provide similar properties, i.e. to the authors knowledge similar promotion effect was obtained with copper.
- In Figure 5 and Table 2, what is the origin of basicity for this catalyst? The authors should explain what is the base sites
Answer: Thank you for this comment. The sentence was added to the manuscript. (Line 293 to 295 in yellow)
- The structure of the obtained catalyst was not shown. The reviewer recommend to draw the model of the catalyst structure after calcination.
Answer: Thank you for this comment. Mixed oxides are obtained after the catalyst calcination.
There is one obtained oxide phase after calcination, corresponding to periclase. However we can note from XRD (Fig. 4) that there is a residual hydrotalcite. This is expressed in the Lines 213-223.

Reviewer 2 Report
1) Equation 1
The authors write the methanation reaction as irreversible. Is that right?
2) What is the role of the product H2O on the catalyst surface behavior?
Author Response
Comments and Suggestions for Authors
- Equation 1 The authors write the methanation reaction as irreversible. Is that right?
Answer: Thank you for this remark. The reaction is an overall reaction, the symbol has been changed in the Eq. 1. (Line 39) in green in the text)
- What is the role of the product H2O on the catalyst surface behavior?
Answer: The support material containing spinel/periclase-like oxides is normally resistant towards increased temperature and water vapor rich environment so situation like structure collapse are not very probable. Although in the post-run XRD the peaks corresponding to periclase or spinel phase were less intense (while Ni0 related peaks were much more intense), which may be explained with the phenomenon of particles non-stabilization. Sentence explaining this behavior was added to the manuscript (1).
- Guil-Lopez, R.; Navarro, R.M.; Ismail, A.A.; Al-Sayari, S.A.; Fierro, J.L.G. Influence of Ni environment on the reactivity of Ni-Mg-Al catalysts for the acetone steam reforming reaction. Int. J. Hydrogen Energy 2015, 40, 5289–5296, doi:10.1016/j.ijhydene.2015.01.159
(Line 343-347)

Reviewer 3 Report
The manuscript is well written, the introduction is argued, the materials are properly characterized and the catalytic activity is corelated withs the catalysts properties. The topic of the study is in good agreement with European Green Deal directions for 2050, where adjustments must be done in the energy sector to reduce GHG emissions, using efficient and sustainable approaches. Therefore, I agree the manuscript for publishing, but there still two observations:
- There is no reference for the Figure 1 in the manuscript. The authors are required to fix this issue and to rename the figures accordingly.
- The broad diffraction lines of the calcined samples (Figure 4a) can be associated with small crystal sizes and /or loss of crystallinity, rather than high crystallinity, as the authors affirm.
Author Response
Comments and Suggestions for Authors
The manuscript is well written, the introduction is argued, the materials are properly characterized and the catalytic activity is corelated withs the catalysts properties. The topic of the study is in good agreement with European Green Deal directions for 2050, where adjustments must be done in the energy sector to reduce GHG emissions, using efficient and sustainable approaches. Therefore, I agree the manuscript for publishing, but there still two observations:
- There is no reference for the Figure 1 in the manuscript. The authors are required to fix this issue and to rename the figures accordingly.
Answer: Thank you for this comment, reference was added and other figures renamed. (Line 113-114 in pink in the text)
- The broad diffraction lines of the calcined samples (Figure 4a) can be associated with small crystal sizes and /or loss of crystallinity, rather than high crystallinity, as the authors affirm.
Answer: Changes were introduced to the text. (Line 223-225)

Reviewer 4 Report
The work entitled "Co-precipitated Ni-Mg-Al hydrotalcite-derived catalyst promoted with vanadium for CO2 methanation" describes interesting results. I would like to recommend the manuscript for publication with minor revision.
The main goal of this study was to demonstrate the promotion effect of V content on textural properties and also the activity of NiMgAl hydrotalcite-derived catalysts for CO2 methanation. Catalysts have been prepared by one-step synthesis via the co-precipitation method. The authors found a correlation between the CO2 conversion, specific surface area and Ni0 crystallite size.
How do the authors explain such significant differences in the surface area, mesopore volume and the size of Ni crystallite when increasing the V content from 1 to 2%? It is interesting that in their previous study [32] they found the same optimal V loading 2% in the case of impregnated NiMgAl hydrotalcite-derived catalysts. Could authors somehow summarize if the method of V loading (impregnation or coprecipitation) influences the textural properties of the catalyst?
There must be a mistake in Table 2 for Ni-CP-V0.0 sample as there is sum of share of crystalline phase equals 110%. Therefore correction should be done also in the text.
In the following, I have technical remarks
What do the authors mean by "reduction of such type of oxides products" on row 64. In the previous sentence the authors mentioned the thermal decomposition of hydrotalcite to mixed oxide, I suppose that the reduction should concern nickel oxide (not MgO or Al2O3) in NiMgAl hydrotalcite derived catalysts by hydrogen. I think this part should be reformulated to be more clear.
r287 : reference [http://dx.doi.org/10.1016/j.apcata.2014.09.028] should be replaced by number
r319: please correct "resuced" to "reduced"
r380 and 386 : CO2 (2 should be in subscript); min-1 (-1 should be in superscript)
Author Response
Comments and Suggestions for Authors
The work entitled "Co-precipitated Ni-Mg-Al hydrotalcite-derived catalyst promoted with vanadium for CO2 methanation" describes interesting results. I would like to recommend the manuscript for publication with minor revision.
The main goal of this study was to demonstrate the promotion effect of V content on textural properties and also the activity of NiMgAl hydrotalcite-derived catalysts for CO2 methanation. Catalysts have been prepared by one-step synthesis via the co-precipitation method. The authors found a correlation between the CO2 conversion, specific surface area and Ni0 crystallite size.
- How do the authors explain such significant differences in the surface area, mesopore volume and the size of Ni crystallite when increasing the V content from 1 to 2%? It is interesting that in their previous study [32] they found the same optimal V loading 2% in the case of impregnated NiMgAl hydrotalcite-derived catalysts. Could authors somehow summarize if the method of V loading (impregnation or coprecipitation) influences the textural properties of the catalyst?
Answer: All the listed properties, such as surface area, mesopore volume and Ni0 crystallite size are corellated. It was found that 2 wt% of vanadium was an optimal promotion leading to a well-developed porosity as well as other properties , such as an increase of specific surface area or the nickel dispersion – without blocking pores with vanadium oxide particles at higher concentrations.
We have added this summary in the conclusion:
Finally, the V-promoted catalysts prepared by co-precipitation exhibited higher basicity and specific surface area, depending on the V-loading. The co-precipitation method resulted in the formation of an uniform hydrotalcite structure, which was later decomposed into mixed oxides during calcination. This homogeneous matrix of mixed oxides appared to lead to a better distribution of nickel on the surface, an enhanced basicity, and a larger specific surface area during the reduction and subsequent reaction. For comparison, nickel oxide was deposited as a separate phase on Mg-Al-V hydrotalcites in the impregnated materials. This resulted in a lower specific surface area, lower basicity (probably due to the blockage of Mg-O sites) and ultimately comparable Ni° crystallite size on the surface..
- There must be a mistake in Table 2 for Ni-CP-V0.0 sample as there is sum of share of crystalline phase equals 110%. Therefore correction should be done also in the text.
Answer: Thank you for this comment, it was an error in copying the data to the table. Correct value was added to table. (Line 264)
In the following, I have technical remarks
- What do the authors mean by "reduction of such type of oxides products" on row 64. In the previous sentence the authors mentioned the thermal decomposition of hydrotalcite to mixed oxide, I suppose that the reduction should concern nickel oxide (not MgO or Al2O3) in NiMgAl hydrotalcite derived catalysts by hydrogen. I think this part should be reformulated to be more clear.
Answer: Thank you for this comment, the sentence was clarified. (Lines 68-70)
- r287 : reference [http://dx.doi.org/10.1016/j.apcata.2014.09.028] should be replaced by number
Answer: The issue was corrected in the text and marked in blue. (Line 332)
- r319: please correct "resuced" to "reduced"
Answer: The issue was corrected in the text and marked in blue. (Line 370)
- r380 and 386 : CO2 (2 should be in subscript); min-1 (-1 should be in superscript)
Answer: The issue was corrected in the text and marked in blue. (Lines 431-438)

Round 2
Reviewer 1 Report
This manuscript is the revised version previously submitted to Molecules.
After careful reading of the response to the reviewer’s comments and the revised manuscript, the reviewer thinks that the manuscript has been sufficiently improved for publication. Therefore, the manuscript is now publishable in Molecules.